# CROSS-ARCHITECTURE KNOWLEDGE DISTILLATION VIA INFORMATION ALIGNMENT

## ABSTRACT

Transformer architectures have demonstrated remarkable success in capturing long-range dependencies and global contextual information, whereas Convolutional Neural Networks (CNNs) remain dominant in many industrial applications due to their efficiency and strong local feature modeling. Bridging the complementary strengths of these architectures, Cross-Architecture Knowledge Distillation (CAKD) has emerged as a promising approach to transfer global knowledge from Transformers to CNNs. However, existing methods either rely on generic distillation strategies that fail to address inductive bias discrepancies, or reduce informative features to logits, which limits generalization across tasks. To overcome these issues, we propose a novel feature-based framework that aligns representations from both structural and semantic perspectives. Structurally, we refine a global information supplement module to extract residual cues through global-local comparison, facilitating more compatible feature transfer. Semantically, we apply the $\ell_1$-regularization to encourage sparse and meaningful global compensation patterns, mimicking Transformer's attention outputs. Extensive experiments on image classification and instance segmentation benchmarks demonstrate that our method effectively mitigates the feature misalignment between Transformers and CNNs, yielding consistent improvements over state-of-the-art works, with up to 2.7% gains on CIFAR-100 and 0.9% on ImageNet-1K, respectively.

## 1 INTRODUCTION

Transformer architecture has achieved remarkable success across diverse deep learning tasks such as computer vision and natural language processing, offering notable advantages over CNNs Bai et al. (2021). With sophisticated self-attention mechanism and positional encoding, Transformer-like models can effectively capture long-range dependencies and richer contextual information, which enables them to achieve significant precision results in laboratory environments. However, in industrial communities, CNNs still occupy more application scenarios with their mature ecosystem, e.g., lower computational load, data requirements, and training costs. Especially in tasks like defect detection and part recognition, where fine-grained local cues are crucial, CNNs are more suitable due to their local feature modeling capacity, compared with Transformers that primarily focus on global information extraction. Therefore, investigating how to leverage the superior capabilities of Transformer-like models to further enhance the performance of CNNs applications holds significant practical importance.

Knowledge distillation (KD), first proposed by Hinton *et al*. Hinton et al. (2015), aims at utilizing the "dark knowledge" stored in a complex teacher model to improve the training of a compact student model. Leveraging knowledge transfer, this paradigm has become one of the most prevalent techniques in model compression, jointly with pruning Yu et al. (2018); Hou et al. (2022) and quantization Liu et al. (2021b); Zhang et al. (2021), and then is extended to other related scenarios such as incremental learning Feng et al. (2022) and federated learning Chen et al. (2023). Therefore, for heterogeneous networks pair, i.e., Transformer teacher and CNN student, researchers have also proposed the method dubbed as Cross-Architecture Knowledge Distillation (CAKD). It constitutes a key technology for meeting the aforementioned demand, thereby reconciling the accuracy of Transformer-based representations with the efficiency of CNN-based inference.

Current methods about CAKD are still in the early stage of exploration. Directly employing general methods without further adaptation cannot adequately bridge the knowledge gap between models with different inductive biases Zhao et al. (2022); Chen et al. (2021). In contrast, existing methods explicitly tailored for CAKD prefer to transform the more informative feature representations into less discriminative logits, so as to achieve more effective knowledge transfer Hao et al. (2023); Li et al. (2024). Nonetheless, such paradigm presents another problem, as its heavy dependence on logits may weaken the generalizability of the distillation to downstream tasks.

To address this issue, we propose a novel feature-based knowledge distillation framework to mitigate the discrepancy between Transformers and CNNs. Focusing on the distinct inductive biases of these two kinds of network architectures, i.e., global relation modeling vs. local information perception, our proposed method tends to solve the feature misalignment in distillation from both structural and semantic aspects. Specifically, we introduce the Global Information Supplement (GIS) module as a modulator to capture the residual representations from global-local comparison, thereby enabling more compatible feature transfer structurally. And then, we impose $\ell_1$-regularization on the global compensation, encouraging them to retain sparse and semantically meaningful patterns akin to those captured by attention mechanism. Based on this, we solve the information asymmetry in CAKD that severely limits the effectiveness of knowledge transfer for Transformers and CNNs.

To evaluate our proposed method, we conducted comprehensive experiments on both image classification and instance segmentation tasks. The final results confirmed our effectiveness in bridging the inductive bias gaps between Transformers and CNNs, i.e., enabling the student models to well inherit global knowledge from the teacher models. The main contributions of this work can be summarized as follows:

- We propose the GIS module, inspired by non-local module, to capture the supplementary cues provided by global information beyond local representations, enabling CNN student to better absorb the knowledge from Transformer-like models.
- We apply $\ell_1$-regularization on supplementary components to imitate the characteristics of attention outputs provided by Transformer teacher, achieving information alignment in semantic aspect.
- Our proposed method achieve the best performance on both image classification and instance segmentation tasks, e.g., outperforming other state-of-the-art methods with gains of up to 2.7% on CIFAR-100 and 0.9% on ImageNet-1K, respectively.

## 2 RELATED WORKS

**Vision Transformer.** Transformer architecture was first proposed by Vaswani *et al* in NLP tasks Vaswani et al. (2017). Leveraging long-range dependencies modeled by attention mechanism, this novel network exhibits a stronger ability to represent global information, leading to higher performance compared to CNNs. Based on this, many studies have been proposed to adapt the Transformer framework to the computer vision domain. As the pioneer, ViT Dosovitskiy et al. (2020) split the images into patches and then mapped them into embedding tokens for subsequent Transformer encoding, just as in NLP tasks. This work achieved the state-of-the-art performance at that time and facilitated the development of subsequent works. Swin-Transformer Liu et al. (2021a) introduced a hierarchical Transformer architecture with shifted windows, enabling scalable modeling of both local and global representations. DeiT Touvron et al. (2021) enhanced Transformer model's training efficiency for vision tasks through knowledge distillation with a dedicated token.

The success of Transformers is built upon quadratic computational complexity and massive data support, which leaves CNNs with strong competitiveness.

**Knowledge Distillation.** KD is an effective approach to employ the supervision of a teacher model for improving the performance of a student model, using logits or features as knowledge carrier. Hinton *et al.* Hinton et al. (2015) established the foundational logits-based framework, while Fit-Net Romero et al. (2014) first introduced feature-based paradigm. Based on this, following works extended the method to various scenarios through model ensembling Zhang et al. (2018); Mirzadeh et al. (2020), contrastive learning Tian et al. (2019), and so on. DKD Zhao et al. (2022) separated target and non-target class knowledge for more effective distillation. DIST Huang et al. (2022) preserved inter- and intra-class prediction relations from a stronger teacher using a correlation-based

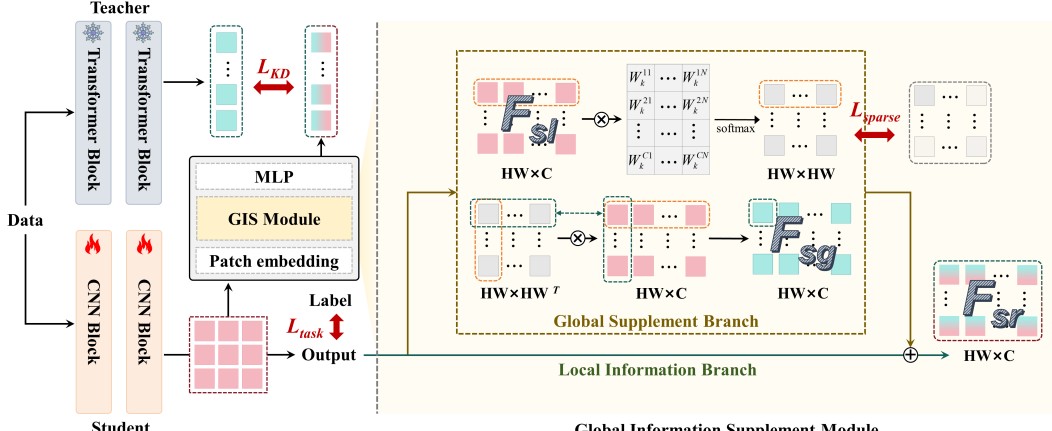

Figure 1: Overview of our proposed feature-based KD method. The GIS module can be divided into two branches for capturing local information and modeling global supplement, respectively.

loss for more effective student training. With regard to knowledge distillation tailored for heterogeneous architectures, existing works are limited. OFA Hao et al. (2023) aligned the features in logits space with multiple exit branches, and adaptively enhanced target information with refined Kullback-Leibler divergence (KL) loss for more effective distillation. TAS Li et al. (2024) introduced an assistant model to bridge heterogeneous teachers and students in distillation, using combined module functions and a spatial-agnostic InfoNCE loss for effective feature alignment.

Existing works about CAKD are accustomed to transform the features in logits space. Purely feature-based approaches remain largely unexplored, making our proposed method feasible.

## 3 METHOD

In this paper, we attribute the bottleneck in cross-architecture knowledge transfer to the overlooking of learning about global knowledge in Transformer-like teacher models. To solve this issue, we propose a novel cross-architecture knowledge distillation method based on an elaborate feature alignment mechanism. The overall pipeline is represented in Figure 1, focusing on the integration of local information from CNN features and global information supplement under sparse regularization.

Before discussing specific feature distillation method, it is necessary to align the feature dimensions of the teacher and student models. Given teacher feature $F_t \in \mathbb{R}^{N \times L \times C}$ and student feature $F_s \in \mathbb{R}^{N \times C \times H \times W}$, both taken from the penultimate layer, conventional approach regards $F_s$ as images and serializes it into patch embeddings directly. However, prior work Kazemnejad et al. (2023) shows that Transformer decoders without positional encoding exhibit stronger generalization than those using Absolute Positional Encoding (APE) Dosovitskiy et al. (2020), Relative Positional Encoding (RPE) Liu et al. (2021a), or other variants. And the serialization here targets CNN features, which already encode spatial information. Therefore, the alignment process mainly serves as a decoding process to adapt them with Transformer features. It can be simplified as

$$F_{sl} = \text{Reshape}(\text{Conv}(F_s)), \tag{1}$$

where Reshape flatten $F_s \in \mathbb{R}^{N \times C \times H \times W}$ into $F_{sl} \in \mathbb{R}^{N \times HW \times C}$. Based on this, the feature distillation loss, i.e., Mean Square Error (MSE) loss, is formulated as

$$\mathcal{L}_{KD} = \frac{1}{N} \sum_{i=1}^{N} \left\| W F_{sl}^i - F_t^i \right\|_F^2, \tag{2}$$

where $N$ denotes the number of samples and $W$ can be a non-linear transformation.

**Global Information Supplement.** Due to limited receptive field, CNNs can only capture local feature information, which hinders them from fully comprehending the knowledge embedded in the

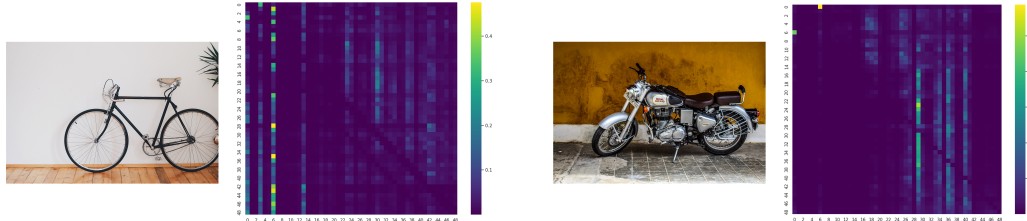

Figure 2: Visualization of the last-layer attention maps of Swin-T.

features of a Transformer teacher during distillation. This issue is further exacerbated by the substantial capacity gap between the teacher and student models, since the mere size of a network does not guarantee its effectiveness as a teacher Huang et al. (2022). Therefore, during the knowledge transfer from Transformer to CNN, modeling the discrepancy between local and global information and selectively distilling reasonable knowledge is essential. Based on this, we propose a Global Information Supplement (GIS) module to facilitate feature distillation. It is inspired from Non-Local module Cao et al. (2019a), a convolutional component for capturing long-range dependencies in the data, and its detailed framework is depicted in Figure 1. GIS employ an attention-like mechanism for modeling supplement information, which can be formulated as

$$F_{sg} = \sigma(W_1 F_{sl}^T) F_{sl}, \tag{3}$$

where $\sigma$ and $\odot$ denote the SoftMax function and Hadamard product, respectively, and $W_1 \in \mathbb{R}^{HW \times C}$ presents a linear transformation. And then, we integrate the supplement feature $F_{sg}$ and the original local feature $F_{sl}$ to generate final distillation target as

$$F_{sr} = W_{\mathrm{mlp}}(F_{sl} + F_{sg}), \tag{4}$$

where $W_{\mathrm{mlp}}$ is a MLP layer, identical to the one used in Transformer block. Obviously, compared to $F_s$, $F_{sr}$ exhibits greater similarity to $F_t$ in terms of information structure. It introduces an additional $F_{sg}$ to fit the discrepancy between global and local information, thereby relaxing the constraints on the student features. Thus, the feature distillation loss is reformulated as

$$\mathcal{L}_{KD} = \frac{1}{N} \sum_{i=1}^{N} \left\| F_{sr}^i - F_t^i \right\|_F^2. \tag{5}$$

**Semantic Information Alignment.** In addition to structural alignment, we also expect the refined feature to be semantically similar to those of the Transformer teacher. By visualizing the last-layer attention maps of Swin-T model (Figure 2), we find that although the softmax-normalized weights are continuous, only a few positions attain high values, resulting in a sparse distribution. To model this property, we impose an $\ell_1$-regularization constraint on $F_{sg}$ to encourage sparsity. This design aligns with the intrinsic attention characteristics of Transformers and further enables the CNN student to approximate global information by aggregating fewer local positions, thereby facilitating the learning of richer global knowledge. The $\ell_1$-regularization is formulated as:

$$\mathcal{L}_{sparse} = \frac{1}{N} \sum_{i=1}^{N} \left\| \sigma(W_1 F_{sl}^T)_i \right\|_1. \tag{6}$$

Therefore, the final loss is the combination of task-specific loss $\mathcal{L}_{task}$, feature distillation loss $\mathcal{L}_{KD}$, and sparsity loss $\mathcal{L}_{sparse}$ as

$$\mathcal{L} = \mathcal{L}_{task} + \beta \mathcal{L}_{KD} + \lambda \mathcal{L}_{sparse}, \tag{7}$$

where $\beta$ is hyperparameter for controlling the strength of distillation and $\lambda = 1e - 4$ as usual.

**Analysis.** Substituting Eq. 3 into Eq. 5, we can easily obtain:

$$\mathcal{L}_{KD} = \left\| (W_{\mathrm{mlp}} F_{sl} - F_t) + W_{\mathrm{mlp}} F_{sg} \right\|^2 \tag{8}$$

$$= \underbrace{\left\| W_{\mathrm{mlp}} F_{sl} - F_t \right\|^2}_{\mathcal{L}_{oriKD}} + \underbrace{2 \langle W_{\mathrm{mlp}} F_{sl} - F_t, W_{\mathrm{mlp}} F_{sg} \rangle}_{\mathcal{L}_{global}} + \underbrace{\left\| W_{\mathrm{mlp}} \sigma(W_1 F_{sl}^T) F_{sl} \right\|^2}_{\mathcal{L}_{extra}}, \tag{9}$$

Table 1: Comparison with state-of-the-art methods on CIFAR-100.

| Teacher Accuracy(%) | Swin-T 89.26 | | ViT-S 92.04 | |
|---|---|---|---|---|
| Student | ResNet-18 | MobileNet-V2 | ResNet-18 | MobileNet-V2 |
| Accuracy(%) | 74.01 | 73.68 | 74.01 | 73.68 |
| KD Hinton et al. (2015) | 78.74 | 74.68 | 77.26 | 72.77 |
| FitNet Romero et al. (2014) | 78.87 | 74.28 | 77.71 | 73.54 |
| CC Peng et al. (2019) | 74.19 | 71.19 | 74.26 | 70.67 |
| RKD Park et al. (2019) | 74.11 | 69.00 | 73.72 | 68.46 |
| CRD Tian et al. (2019) | 77.63 | 79.80 | 76.60 | 78.14 |
| DKD Zhao et al. (2022) | 80.26 | 71.07 | 78.10 | 69.80 |
| DIST Huang et al. (2022) | 77.75 | 72.89 | 76.49 | 72.54 |
| OFA Hao et al. (2023) | 80.54 | 80.98 | 80.15 | 78.45 |
| TAS Li et al. (2024) | 81.61 | 81.28 | 81.93 | **82.10** |
| **Ours** | **84.31** | **83.24** | **82.84** | 81.69 |

where we omit the sum operation for loss and divide $\mathcal{L}_{KD}$ into three terms. $\mathcal{L}_{oriKD}$ is the original KD loss, aiming to force $F_{sl}$ to approach $F_t$ directly, while $\mathcal{L}_{global}$ is our proposed global supplement loss for filling the discrepancy between $F_{sl}$ and $F_{sg}$. These two loss terms form a system akin to an elastic mechanism, which can flexibly and adaptively modulate the student's learning strength from the knowledge in teacher features. However, the extra loss term $\mathcal{L}_{extra}$ tends to weaken the implicit global information of the student features. While the SoftMax function facilitates the construction of long-range dependencies in $F_{sl}$, $\mathcal{L}_{extra}$ drives its outputs toward excessive uniformity. Therefore, we propose to use $\ell_1$-regularization as Eq. 6 to prevent this undesired effect and highlight the informative positions in $F_{sl}$.

## 4 EXPERIMENTS

We conducted comprehensive experiments to evaluate the effectiveness of our proposed method. In this section, we will first report the main results on two representative tasks, i.e., image classification and instance segmentation, and then present the ablation studies and extend experiments for providing more insights of our method. All experiments are trained and validated on 4xNVIDIA RTX 4090D GPUs, based on the Pytorch deep learning framework Paszke et al. (1912). The codebase is provided by the open-source framework of Hao et al. (2023) and Chen et al. (2019), and will be released soon.

### 4.1 RESULTS ON IMAGE CLASSIFICATION

We first verified our method on CIFAR-100 and ImageNet-1K, two benchmark datasets of image classification task, and compared it with other current advanced knowledge distillation methods, including KD Hinton et al. (2015), FitNet Romero et al. (2014), CC Peng et al. (2019), RKD Park et al. (2019), CRD Tian et al. (2019), DKD Zhao et al. (2022), DIST Huang et al. (2022), OFA Hao et al. (2023), and TAS Li et al. (2024).

**Datasets.** CIFAR-100 Krizhevsky et al. (2009) is a benchmark collection for visual recognition, comprising 60,000 natural images distributed across 100 fine-grained classes. It contains 50,000 training and 10,000 testing images, serving as a challenging extension to CIFAR-10. ImageNet Russakovsky et al. (2015) is a large-scale benchmark dataset in computer vision, containing approximately 1.28 million training images and 50,000 validation images, meticulously labeled across 1,000 object categories.

**Models.** For comprehensive analysis, we evaluate our method on diverse network architectures. We choose Swin-T Liu et al. (2021a), DeiT Touvron et al. (2021), and ViT-S Dosovitskiy et al. (2020) as

Table 2: Comparison with state-of-the-art methods on ImageNet-1K.

| Teacher
Accuracy(%) | Swin-T
81.38 | | DeiT-T
72.17 | |
|---|---|---|---|---|
| Student
Accuracy(%) | ResNet18
69.75 | MobileNetV2
68.87 | ResNet18
69.75 | MobileNetV2
68.87 |
| KD Hinton et al. (2015) | 71.14 | 72.05 | 70.22 | 70.87 |
| FitNet Romero et al. (2014) | 71.18 | 71.75 | 70.44 | 70.95 |
| CC Peng et al. (2019) | 70.07 | 70.69 | 69.77 | 70.69 |
| RKD Park et al. (2019) | 68.89 | 67.52 | 69.47 | 69.72 |
| CRD Tian et al. (2019) | 69.09 | 69.58 | 69.25 | 69.6 |
| DKD Zhao et al. (2022) | 71.10 | 71.71 | 69.39 | 70.14 |
| DIST Huang et al. (2022) | 70.91 | 71.76 | 70.64 | 71.08 |
| OFA Hao et al. (2023) | 71.76 | 72.32 | 71.01 | 71.39 |
| TAS Li et al. (2024) | 72.21 | 72.54 | 71.22 | 71.78 |
| **Ours** | **72.27** | **73.45** | **72.10** | **72.61** |

Transformer-like teacher models to distill CNN-like student models of ResNet-18 He et al. (2016) and MobileNet-V2 Sandler et al. (2018).

**Implementation Details.** On CIFAR-100, input images are resized to 224×224 to ensure consistency between teacher and student outputs. All models are trained for 300 epochs with a batch size of 1024, using SGD with momentum 0.9 and weight decay of 2e-3. A cosine learning rate schedule is adopted, decaying from an initial value of 0.1 to a minimum of 0.001. The distillation loss weight is linearly warmed-up during the first 20 epochs. While on ImageNet-1K, training is conducted for 100 epochs with a batch size of 1024, employing SGD with momentum 0.9 and weight decay of 1e-4. The cosine learning rate schedule is applied in the same manner, decaying from 0.1 to 0.001. The warm-up of distillation loss weight is conducted over the first 3 epochs.

**Results.** We summarize the Top-1 accuracy of listed methods on CIFAR-100 and ImageNet-1K in Table 1 and Table 2, respectively. For CIFAR-100, our proposed method markedly improves student performance, leading to an average gain of 9.03% across four settings and achieving comparable or even better performance. With Swin-T as the teacher, Top-1 accuracy increases by 10.30% on ResNet18 (74.01% to 84.31%) and 9.56% on MobileNet-V2 (73.68% to 83.24%). With ViT-S as the teacher, the improvements are 8.23% and 8.01%, respectively. Similarly, our method also achieves clear gains on ImageNet-1K, leading to an average gain of 3.30% across four settings. With Swin-T as the teacher, Top-1 accuracy increases from 69.75% to 72.27% on ResNet18 and from 68.87% to 73.45% on MobileNet-V2. With DeiT-T as the teacher, the improvements are 2.35% and 3.74%, respectively. Compared with representative methods tailored for CAKD, i.e., OFA and TAS, our approach consistently achieves higher accuracy, with particularly notable improvements on MobileNet-V2.

## 4.2 RESULTS ON INSTANCE SEGMENTATION

To further evaluate the generalization of our feature distillation method, we also conducted instance segmentation experiments on MS-COCO using Mask R-CNN, where the output features of FPN are distilled across three teacher-student pairs, i.e., Swin-T as teacher and ResNet-18, ResNet-50, MobileNet-V2 as students.

**Datasets.** MS-COCO Lin et al. (2014) is a large-scale benchmark widely adopted in computer vision. It consists of over 330,000 images depicting complex everyday scenes, with more than 200,000 labeled instances across 80 object categories. Unlike earlier datasets, COCO provides rich and diverse annotations including bounding boxes and segmentation masks, making it a comprehensive resource for evaluating and advancing models in object detection and segmentation.

**Models.** Mask R-CNN He et al. (2017) is a two-stage framework that extends Faster R-CNN Ren et al. (2015) by adding a parallel branch for pixel-level mask prediction. When combined with FPN Lin et al. (2017), it achieves improved accuracy through multi-scale feature representations.

Table 3: Results of instance segmentation on MS-COCO 2017.

| | | mAP | $AP_0$ | $AP_b$ | $AP_t$ | $AP_u$ | $AP_s$ |
|---|---|---|---|---|---|---|---|
| Swin-T | | 39.8 | 63.3 | 42.7 | 53.6 | 43.1 | 24.2 |
| MV2-FPN | w/o KD | 28.3 | 47.2 | 29.9 | 41.7 | 29.0 | 12.1 |
| | Ours | **31.1** (+2.8) | **50.3** | **33.2** | **46.8** | **31.9** | **13.1** |
| R18-FPN | w/o KD | 31.1 | 51.0 | 33.1 | 45.5 | 32.8 | 14.2 |
| | Ours | **33.2** (+2.1) | **53.0** | **34.9** | **48.1** | **34.7** | **14.6** |
| R50-FPN | w/o KD | 34.7 | 55.7 | 37.2 | 47.2 | 37.4 | 18.3 |
| | Ours | **36.3** (+1.6) | **57.2** | **38.8** | **52.5** | **38.6** | **18.6** |

**Implementation Details.** All teacher–student pairs are trained for 12 epochs with an initial learning rate of 0.02, reduced by a factor of 0.1 at the 8th and 11th epochs. Training uses a batch size of 16 and SGD with momentum 0.9 and weight decay 1e-4. Model performance is evaluated using mAP, $AP_{50}$, $AP_{75}$, $AP_S$, $AP_M$, and $AP_L$.

**Results.** As shown in Table 3, the proposed method consistently improves student network performance across different teacher-student setups. The mAP of ResNet-18 increases from 31.1 to 33.2 (+6.8%), and that of ResNet-50 rises from 34.7 to 36.3 (+4.6%). The most pronounced gain is on MobileNet-V2, with mAP improving from 28.3 to 31.1 (+9.9%). Metrics such as $AP_{50}$, $AP_{75}$ also show substantial improvements, further demonstrating the effectiveness and transferability of our method to dense prediction tasks.

Table 4: Ablation results on CIFAR-100 for component analysis.

| | | | | | Swin-T | | ViT-S | |
|---|---|---|---|---|---|---|---|---|
| KD | PE | MLP | GIS | $\mathcal{L}_{sparse}$ | ResNet-18 | MobileNet-V2 | ResNet-18 | MobileNet-V2 |
| ✗ | ✗ | ✗ | ✗ | ✗ | 74.01 | 73.68 | 74.01 | 73.68 |
| ✓ | ✗ | ✗ | ✗ | ✗ | 79.56 | 80.11 | 78.28 | 78.12 |
| ✓ | ✓ | ✗ | ✗ | ✗ | 82.89 | 81.23 | 81.68 | 79.70 |
| ✓ | ✓ | ✓ | ✗ | ✗ | 83.26 | 81.86 | 82.07 | 79.51 |
| ✓ | ✓ | ✓ | ✓ | ✗ | 83.83 | 82.32 | 82.31 | 81.45 |
| ✓ | ✓ | ✓ | ✓ | ✓ | **84.31** | **83.24** | **82.84** | **81.69** |

### 4.3 EXTEND EXPERIMENTS

#### 4.3.1 ABLATION STUDIES

For a more comprehensive exploration of our proposed method, we also conducted ablation studies from components analysis, encoding type, and module architecture three aspects on CIFAR-100.

**Component Analysis.** We conducted ablation studies to evaluate the contribution of each component in our method and presented the results in Table 4, where optimal and suboptimal results are highlighted in bold and underlined, respectively. The baseline just employ a convolution layer as the projector to align the features of teacher and student and "PE" denotes the operation of patch embedding. The ablation results show that simultaneously adopting the proposed GIS module and the regularization $\mathcal{L}_{sparse}$ can achieve the best performance, indicating that all the components in our method are reasonable and effective.

**Encoding Type.** We hypothesize that explicit position encoding is unnecessary in cross-architecture distillation, since CNN features inherently encode spatial relations and size adjustment can be re-

Table 5: Ablation results on CIFAR-100 for encoding type and module architecture.

| Teacher | Swin-T | | | | Teacher | Swin-T | | | |
|---|---|---|---|---|---|---|---|---|---|
| Student | ResNet18 | | MobileNetV2 | | Student | ResNet18 | | MobileNetV2 | |
| Acc(%) | Top-1 | Top-5 | Top-1 | Top-5 | Acc(%) | Top-1 | Top-5 | Top-1 | Top-5 |
| APE | 83.06 | 96.54 | 82.13 | 96.23 | NL | 83.45 | 96.33 | 82.85 | 96.28 |
| RPE | 83.60 | 96.59 | 82.57 | 96.42 | S-NL | 83.63 | 96.31 | 82.73 | 96.51 |
| **Ours** | **84.31** | **96.72** | **83.24** | **96.64** | **Ours** | **84.31** | **96.72** | **83.24** | **96.64** |

garded as decoding this information. To validate this, we compare our design without position encoding against absolute (APE) and relative (RPE) schemes, using Swin-T as the teacher and ResNet-18/MobileNet-V2 as students. As shown in Table 5 (left), the non-explicit design consistently achieves higher Top-1 and Top-5 accuracy, confirming that CNN features already contain sufficient positional information and that additional encoding is redundant.

**Module Architecture.** We also simplify the Non-local module by directly weighting integrated feature attention to facilitate global information learning. Compared with the original and simplified Non-local modules (NL Cao et al. (2019a) and S-NL Cao et al. (2019b)), our design consistently achieves higher Top-1 and Top-5 accuracy as shown in Table 5 (right). These results confirm that the proposed approach enables CNNs to capture global context more effectively, leading to smoother knowledge transfer and improved performance.

**Pretrained Transferability.** To assess the transferability of the knowledge extracted from large-scale pretrained Transformers, we distill the DINOv2 Oquab et al. (2023) into ResNet-18 and MobileNet-V2 on ImageNet-1K, and then employ them as new pretrained models for instance segmentation on MS-COCO using the Mask R-CNN framework. The results demonstrate consistent improvements (e.g., +1.5 mAP for MobileNet-V2 and +0.9 mAP for ResNet-18), particularly on medium and large objects, while also reducing training cost. These findings confirm the effectiveness of our method in enabling efficient cross-architecture transfer of pretrained feature knowledge.

Table 6: Results of instance segmentation for pre-training transfer on MS-COCO 2017.

| | | mAP | $AP_0$ | $AP_b$ | $AP_t$ | $AP_u$ | $AP_s$ |
|---|---|---|---|---|---|---|---|
| MV2-FPN | w/o distill | 28.3 | 47.2 | 29.9 | 41.7 | 29.0 | 12.1 |
| | DINOv2 | **29.8** (+1.5) | **49.8** | **32.7** | **45.4** | **31.3** | **12.6** |
| R18-FPN | w/o distill | 31.1 | 51.0 | 33.1 | 45.5 | 32.8 | 14.2 |
| | DINOv2 | **32.0** (+0.9) | **51.8** | **33.9** | **46.8** | **33.9** | **14.4** |

### 4.3.2 VISUAL ANALYSIS

The visualization results on CIFAR-100, adopting Swin-T as teacher and ResNet-18 as student, provide intuitive evidence of the effectiveness of our proposed distillation method as shown in Figure 4.3.2.

**Visualization of t-SNE.** At logits-level, the results of t-SNE show that the original student model exhibits dispersed and overlapping clusters, while the teacher model forms compact and well-separated clusters. After distillation of our method, the student clusters become clearer and more discriminative, indicating a stronger ability to distinguish categories.

**Visualization of Attention Maps.** At feature-level, the visualization of attention maps demonstrates the effectiveness of our GIS module. The attention patterns obtained form distilled ResNet-18 resemble the Swin-T Transformer's last-layer attention, confirming the transfer of global knowledge. In pixel-visual analysis, the central positions of maps present more reliance on local attention due to larger receptive fields, whereas edge and corner positions expand their attention ranges to compensate for limited receptive fields. This pattern aligns with the design principle of GIS module, validating its role in bridging CNN's local features with Transformer's global information.

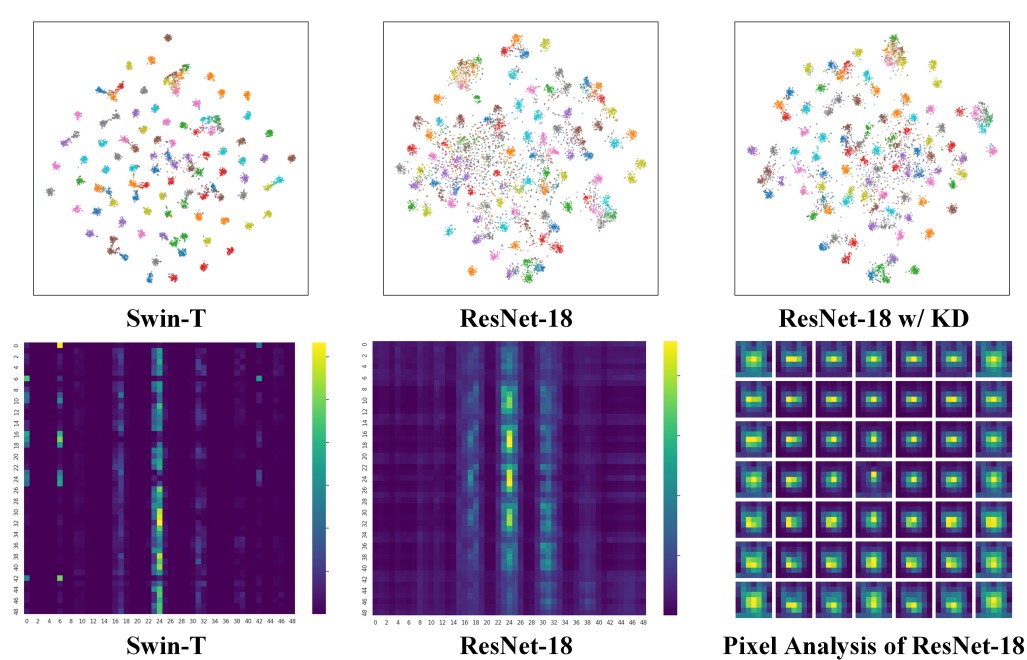

Figure 3: **Visualization of t-SNE and pixel analysis.** We visualized the model's logits (upper) and features (bottom) separately to intuitively demonstrate the effectiveness of our method.

## 5 CONCLUSION

In this work, we addressed the critical challenge of bridging the knowledge gap between Transformer teachers and CNN students in cross-architecture knowledge distillation. Unlike prior works that either overlook inductive bias discrepancies or overly rely on logits-based distillation, we proposed a novel feature-based distillation framework that explicitly aligns global–local representations from both structural and semantic perspectives. By redesigning the non-local module into GIS to capture residual cues and applying $\ell_1$-regularization to enforce sparsity, our method effectively mitigates feature misalignment and enhances knowledge transfer. Extensive experiments on image classification and instance segmentation tasks validated the effectiveness of our proposed method, achieving consistent improvements over other state-of-the-art methods. Moving forward, we plan to extend our framework to broader application scenarios, such as multimodal learning and large-scale pre-trained models, further exploring its potential in enhancing the practicality and versatility of knowledge distillation.

## ETHICS STATEMENT

We declare that this work was conducted in accordance with the ICLR Code of Ethics. No part of this study involved experiments on humans, animals, or sensitive personal data. All datasets used in this paper are publicly available and employed strictly for research purposes.

## REPRODUCIBILITY STATEMENT

We declare that all experimental results in this work are reproducible. All implementation details, including network architectures, training settings, hyperparameters, and evaluation protocols, are described in the paper. The source code and trained models will be released upon publication to facilitate independent reproduction and future research.

## USAGE OF LLMS

In this paper, we employ LLMs solely to provide translation suggestions and polish the manuscript, without any other operations.

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
