# OpenReview forum: "Cross-Architecture Knowledge Distillation via Information Alignment"
_ICLR.cc/2026/Conference — ICLR 2026 Conference Withdrawn Submission_

### Official Review · Reviewer_7XXQ · 2025-10-31

**Soundness:** 3
**Presentation:** 3
**Contribution:** 3
**Rating:** 4
**Confidence:** 4

**Summary:**

This paper proposes a feature-based knowledge distillation method to transfer knowledge from Transformer-based models to CNN-based models. It consists of two components: (1) a global information supplement module is introduced to enhance global information (in CNN knowledge) for heterogeneous distillation; and (2) an additional sparsity term is added to the constructed attention map to mimic the attention map of Transformer-based models. Experiments on image and instance segmentation tasks demonstrate the effectiveness of the proposed method.

**Strengths:**

- The paper is generally well-written
- Experiments on multiple tasks are conducted.
- The method part has a lot of information.

**Weaknesses:**

- The current proposed scheme only leverages the Transformer-based teachers to supervise CNN-based students. Can the proposed method be used to transfer knowledge from CNN-based models to Transformer-based ones, similar to the referred methods OFA and TAS?

- Compared with distilling knowledge in the logit space (e.g., OFA), does the proposed method require more memory and computational overhead in training phase? Additionally, would integrating the proposed method with logit-based KD (e.g., OFA) result in better outcomes?

- In Eq. (3), there is no $\odot$. Moreover, $ F_{sl}^T $ should be ${ F_{sl}^i}^{T} $. In Eq. (5), $ F_{sl}^i \in {{\mathbb{R}}^{HW \times C}} $ and $ F_{t}^i \in {{\mathbb{R}}^{L \times C}} $, leading to a dimension mismatch.

- In Table 4, to demonstrate the effectiveness of the added $F_{sg}$ (Eq. (3)), the results of 'KD' and 'KD+GIS' should be compared.

- More recent studies should be referred, such as [1,2,3,4].

    [1] Liu Y, Cao J, Li B, et al. Cross-architecture knowledge distillation[C]. ACCV 2022. \
    [2] Zhang W, Liu Y, Ran W, et al. Cross-Architecture Distillation Made Simple with Redundancy Suppression[C]. ICCV 2025. \
    [3] Xu L, Liu K, Liu J, et al. Local Dense Logit Relations for Enhanced Knowledge Distillation[C]. ICCV 2025. \
    [4] Lin J H, Yao Y, Hsu C F, et al. Perspective-Aware Teaching: Adapting Knowledge for Heterogeneous Distillation[C]. ICCV 2025.

**Questions:**

Please see the weaknesses.

---

> ### Author Response · Authors · 2025-11-21
> **Response to Reviewer 7XXQ**
>
> **Response to W1:** Following the experimental setup of OFA, we conducted experiments on CIFAR-100 to distill CNNs into Transformer-like models, where we set β=1 and λ=1e-5 without detailed tuning. As shown by the accuracy results in the table below, our method achieves significant performance improvements across both combinations. This demonstrates that our approach facilitates effective knowledge transfer between CNNs and Transformer-like models, regardless of the direction of distillation. And we will add these results into our paper.
>
> | Student       | Teacher | OFA   | RSD[1] | LDRLD[2] | PAT[3] | Ours  |
> |---------------|---------|-------|--------|----------|--------|-------|
> | ConvNeXt-T    | Swin-P  | 78.32 | 82.21  | 80.71    | 80.74  | **82.41** |
> | ConvNeXt-T    | DeiT-T  | 75.76 | 82.46  | 77.46    | 79.59  | **83.09** |
>
> ---
>
> **Response to W2.1:** For an intuitive view of our method’s complexity, we compare the extra parameters and total distillation FLOPs introduced by our approach with those of OFA. The results are summarized as follows. Obviously, both the extra parameters and overall FLOPs of our method are lower than those of OFA.
>
> | Dataset | Swin-T Res-18 | ViT-S Res-18 | Swin-T MV-2 | ViT-S MV-2 | Swin-T Res-18 | DeiT-T Res-18 | Swin-T MV-2 | DeiT-T MV-2 |
> |---------|----------------|--------------|--------------|-------------|----------------|----------------|--------------|--------------|
> |        | CIFAR-100  |              |              |             | ImageNet-1K|                |              |              |
> | FLOPs of OURs  | 6281.03 M | 6169.07 M | 4764.37 M | 4648.79 M | 6282.18 M | 2931.38 M | 4766.21 M | 1409.99 M |
> | Extra Params of OURs | 1.76M | 0.53M | 1.91M | 0.61M | 1.76M | 0.16M | 1.91M | 0.20M |
> | FLOPs of OFA | 6286.19 M | 6147.11 M | 4774.43 M | 4651.71 M | 6290.11 M | 2979.61 M | 4780.88 M | 1487.66 M |
> | Extra Params of OFA | 1.63M | 1.19M | 2.53M | 2.53M | 4.39M | 3.04M | 7.14M | 7.14M |
> | Params of S | 11.23M | 11.23M | 2.35M | 2.35M | 11.69M | 11.69M | 3.50M | 3.50M |
>
> ---
>
> **Response to W2.2:** Yes, in general, combining feature-based KD with logits-based KD can further boost performance. Building upon our method, we additionally incorporate a KL divergence loss between the logits, and the resulting accuracy is shown below (\bate=30, λ=1e-4, kl=10). However, considering that feature distillation may be directly applied to other downstream tasks, we did not include an additional logits-based KL loss, as it inherently depends on the classifier.
>
> | Method         | w/o KD | w/ KD |
> |----------------|--------|--------|
> | Swin-T Res-18  | 84.31  | 84.81  |
>
> ---
>
> **Response to W3:** Thank you for pointing out the errors. We will recheck the symbols and equations in our paper. In Eq.3, since the operation between the two matrices is matrix multiplication, no Hadamard product symbol is used. And F_sl is transformed from HWxC into LxC by W. We apologize for not specifying the above details in the paper, which may have caused the misunderstanding.
>
> ---
>
> **Response to W4:** Since we believe that the potential role of the MLP is to provide nonlinear transformation and residual link for conventional feature distillation methods (KD in Table 4), we consider this essential in distillation and therefore treat it as a basic operation for comparison.
>
> ---
>
> **Response to W5:** Really thanks for providing these cutting-edge methods for comparison with our method. This significantly improves the responsiveness of our method. We will access these works in our paper.
>
> | Method | C100: Swin-T Res-18 | C100: ViT-S Res-18 | C100: Swin-T MV-2 | C100: ViT-S MV-2 | IN1K: Swin-T Res-18 | IN1K: DeiT-T Res-18 | IN1K: Swin-T MV-2 | IN1K: DeiT-T MV-2 |
> |--------|----------------------|---------------------|---------------------|--------------------|-----------------------|----------------------|---------------------|----------------------|
> | RSD[1]   | 83.92 | 81.50 | **83.68** | 81.68 | 72.13 | 71.70 | 72.36 | 72.18 |
> | LDRLD[2] | 82.17 | 80.36 | 81.64 | 79.21 | -     | -     | -     | -     |
> | PAT[3]   | 81.22 | 80.11 | 78.78 | 78.87 | 71.54 | -     | -     | -     |
> | Ours     | **84.31** | **82.84** | 83.24 | **81.69** | **72.27** | **72.10** | **73.45** | **72.61** |
>
> [1] Zhang W, Liu Y, Ran W, et al. Cross-Architecture Distillation Made Simple with Redundancy Suppression[C]. ICCV 2025.
>
> [2] Xu L, Liu K, Liu J, et al. Local Dense Logit Relations for Enhanced Knowledge Distillation[C]. ICCV 2025.
>
> [3] Lin J H, Yao Y, Hsu C F, et al. Perspective-Aware Teaching: Adapting Knowledge for Heterogeneous Distillation[C]. ICCV 2025.
>
> [4] Liu Y, Cao J, Li B, et al. Cross-architecture knowledge distillation[C]. ACCV 2022.

---

### Official Review · Reviewer_hshi · 2025-10-31

**Soundness:** 3
**Presentation:** 3
**Contribution:** 3
**Rating:** 6
**Confidence:** 3

**Summary:**

This paper targets the cross-architecture knowledge distillation (CAKD) setting, where a Transformer teacher and a CNN student exhibit a “global vs. local” inductive-bias mismatch, and proposes a feature-level distillation framework, The core idea is to insert a Global Information Supplement (GIS) module on the student side: it aggregates a “global compensation” from the student’s local features via an attention-style mechanism, fuses this with the original features, and then aligns the result to the teacher’s intermediate features using MSE. In parallel, an ℓ1 sparsity regularization is applied to the aggregation weights to make them resemble the sparse pattern commonly observed in Transformer attention. The authors further find that no explicit positional encoding is required to achieve better alignment. Experiments on CIFAR-100 and ImageNet-1K and COCO show consistent gains over generic KD and several CAKD baselines, and ablations plus visualizations (t-SNE, attention maps) validate each component. The added modules and losses are training-only, so inference incurs no extra overhead, yielding good practical utility.

**Strengths:**

1.The proposed method is simple and elegant, but significantly effective.
2.The observation that does not require explicit PE is enlightening.

**Weaknesses:**

1.Some expressions in the article require careful consideration. For example, 133-134"Purely feature-based approaches remain largely unexplored". The following articles all conducted explorations from the perspective of features："Cross-Architecture Distillation Made Simple with Redundancy Suppression","Cross-Architecture Knowledge Distillation" and so on. Also,138-139"In this paper, we attribute the bottleneck in cross-architecture knowledge transfer to the overlooking of learning about global knowledge in Transformer-like teacher models." is quite ambiguous, you may point out it's student’s insufficient learning of the teacher’s global knowledge.
2.During training, the method introduces the GIS module and a sparsity regularizer and requires access to the teacher’s intermediate features. The training resource threshold and memory overhead remain unquantified.

**Questions:**

1.The comparison methods mentioned in the article only cover up to 2024. Have you compared the methods for the first half of 2025?
2.Is using only the penultimate layer optimal? Have you explored the benefits and costs of multi-layer or hierarchical distillation?

---

> ### Author Response · Authors · 2025-11-21
> **Response to hshi**
>
> **Response to W1:** We thank the reviewer for this suggestion. We have revised the manuscript: the first statement now emphasizes that feature-based cross-architecture distillation remains a wide scope for exploration, and the second clarifies that the bottleneck is the student’s insufficient learning of the teacher’s global knowledge.
>
> ---
>
> **Response to W2:** Since the GIS also serves to align the feature dimensions between the teacher and student models, its extra parameters and computational cost largely depend on the architectures of them. To provide a clearer view of the method’s complexity, we compare the extra parameters and total distillation FLOPs introduced by our approach with those of OFA. The results are summarized in the table below. Notably, both the extra parameters and overall FLOPs of our method are comparable to—or even lower than—those of OFA.
>
> | Dataset | Swin-T Res-18 | ViT-S Res-18 | Swin-T MV-2 | ViT-S MV-2 | Swin-T Res-18 | DeiT-T Res-18 | Swin-T MV-2 | DeiT-T MV-2 |
> |---------|----------------|--------------|--------------|-------------|----------------|----------------|--------------|--------------|
> |        | CIFAR-100  |              |              |             | ImageNet-1K|                |              |              |
> | FLOPs of OURs  | 6281.03 M | 6169.07 M | 4764.37 M | 4648.79 M | 6282.18 M | 2931.38 M | 4766.21 M | 1409.99 M |
> | Extra Params of OURs | 1.76M | 0.53M | 1.91M | 0.61M | 1.76M | 0.16M | 1.91M | 0.20M |
> | FLOPs of OFA | 6286.19 M | 6147.11 M | 4774.43 M | 4651.71 M | 6290.11 M | 2979.61 M | 4780.88 M | 1487.66 M |
> | Extra Params of OFA | 1.63M | 1.19M | 2.53M | 2.53M | 4.39M | 3.04M | 7.14M | 7.14M |
> | Params of S | 11.23M | 11.23M | 2.35M | 2.35M | 11.69M | 11.69M | 3.50M | 3.50M |
>
> ---
>
> **Response to Q1:** Thanks to the Reviewer 7XXQ who provides us with multiple cutting-edge works in ICCV 2025, and we compare their results with our method as follows.
>
> | Method | C100: Swin-T Res-18 | C100: ViT-S Res-18 | C100: Swin-T MV-2 | C100: ViT-S MV-2 | IN1K: Swin-T Res-18 | IN1K: DeiT-T Res-18 | IN1K: Swin-T MV-2 | IN1K: DeiT-T MV-2 |
> |--------|----------------------|---------------------|---------------------|--------------------|-----------------------|----------------------|---------------------|----------------------|
> | RSD[1]   | 83.92 | 81.50 | **83.68** | 81.68 | 72.13 | 71.70 | 72.36 | 72.18 |
> | LDRLD[2] | 82.17 | 80.36 | 81.64 | 79.21 | -     | -     | -     | -     |
> | PAT[3]   | 81.22 | 80.11 | 78.78 | 78.87 | 71.54 | -     | -     | -     |
> | Ours     | **84.31** | **82.84** | 83.24 | **81.69** | **72.27** | **72.10** | **73.45** | **72.61** |
>
> [1] Zhang W, Liu Y, Ran W, et al. Cross-Architecture Distillation Made Simple with Redundancy Suppression[C]. ICCV 2025.
>
> [2] Xu L, Liu K, Liu J, et al. Local Dense Logit Relations for Enhanced Knowledge Distillation[C]. ICCV 2025.
>
> [3] Lin J H, Yao Y, Hsu C F, et al. Perspective-Aware Teaching: Adapting Knowledge for Heterogeneous Distillation[C]. ICCV 2025.
>
> ---
>
> **Response to Q2:** Drawing on our previous experience, we find that for image classification such a simple task, distilling the penultimate features (i.e., those before the avgpool layer) is generally the most effective strategy. As shown in the table, with the same hyperparameter settings, distilling intermediate features provides weaker supervision, whereas multi-layer distillation adds extra parameters without yielding clear benefits. Therefore, we focus solely on the penultimate features rather than incorporating additional layers, since our method is not specifically tailored for multi-layer feature distillation like ReviewKD. But for more complex tasks such as object detection and segmentation, processing the output features of backbone maybe better.
>
> | Student | Intermediate Layer | Penultimate Layer | Multi-Layer |
> |---------|--------------|-------------|------------|
> | 74.01   | 76.89        | 84.31       | 84.23      |

---

### Official Review · Reviewer_ZMxK · 2025-11-01

**Soundness:** 3
**Presentation:** 2
**Contribution:** 2
**Rating:** 6
**Confidence:** 4

**Summary:**

In this paper, the authors propose a knowledge distillation approach to distill a Transformer model into a CNN model. The authors first present the related works of vision transofrmer and knowledge disttilation. Then, they propose their method, including a global information supplement method and a semantic information alignment. In addition, they analyze the loss function of the global information supplement method. Finally, they conducted an experimentation to show the advantages of the proposed approach. The idea of the global information method and semantic information alignment is interesting. In addition, the experimental results demonstrate the advantages of the proposed approach.

**Strengths:**

1. The proposed approach, including the global information supplement method and the semantic information alignment is interesting and appears to be effective in the knowledge distillation.
2. The experimentation results show clear advantages of the proposed approach.
3. The analysis of the global information provides additional breakdown of the global information supplement method.

**Weaknesses:**

1. The experimental results exploit two datasets, i.e., CIFAR-100 and ImageNet. More datasets can be exploited to show the generalization of the proposed approach.
2. The structure of the paper can be improved, e.g., the analysis section can be moved before semantic information alignment.
3. Theoretical analysis of the proposed approach can be provided to show the convergence.
4. The """ in "”PE”" can be adjusted.

**Questions:**

1. How to adjust the hyperparameters, e.g., β.
2. Are W and W1 trained during the knowledge distillation process?
3. Is there any theoretical analysis for the convergence of the proposed approach?

---

> ### Author Response · Authors · 2025-11-21
> **Response to Reviewer ZMxK**
>
> **Response to W1:** To assess the generalizability of our method, we additionally performed experiments on the MS-COCO datasets in paper. Nevertheless, as most existing general knowledge distillation methods are evaluated on classification tasks, the primary experimental validation in this work remains focused on CIFAR-100 and ImageNet-1K two benchmarks.
>
> ---
>
> **Response to W2:** Indeed, that's the case. We also agree that moving the analysis before the introduction of our method can make the logic more coherent. Really thanks for your thoughtful suggestion.
>
> ---
>
> **Response to W3:** According to Eq.7, both L_task (Cross Entropy Loss) and L_KD (MSE Loss) are convex functions for input student features/logits. And while the ℓ1 norm is not strictly differentiable, PyTorch provides a subgradient-based approximation for its optimization. Therefore, the overall loss L is semi-algebraic and satisfies Kurdyka–Lojasiewicz rule, ensuring the convergence of subgradient descent.
>
> ---
>
> **Response to W4:** Sincerely apologize for the typos errors in the paper. We will carefully recheck the manuscript.
>
> ---
>
> **Response to Q1:** We conducted grid search for kd loss weight β and the results are summarized as below. For regularization coefficient λ, we just simply set it to 1e-4.
>
> | k5   | k10  | k15  | k20  | k25  | k30  | k35  |
> |------|------|------|------|------|------|------|
> | 82.12 | 83.62 | 83.97 | 84.11 | 84.27 | 84.31 | 84.27 |
>
> For example of Swin-T & Res-18, the performance becomes stable once β exceeds a certain range. Therefore, we set β=30 for distilling Swin-T models in our paper, while for other settings such as the reverse-direction distillation experiment from a CNN to a Transformer-like model that we will include (as below), β is set to 1.
>
>
> | Student       | Teacher | OFA   | RSD | LDRLD | PAT | Ours  |
> |---------------|---------|-------|--------|----------|--------|-------|
> | ConvNeXt-T    | Swin-P  | 78.32 | 82.21  | 80.71    | 80.74  | **82.41** |
> | ConvNeXt-T    | DeiT-T  | 75.76 | 82.46  | 77.46    | 79.59  | **83.09** |
>
> ---
>
> **Response to Q2:** Yes, all modules introduced by our method are trained during distillation. They are both bottleneck structures for avoiding massive parameters.
>
> ---
>
> **Response to Q3:** Please refer to Response to W3.

---

### Official Review · Reviewer_i5NX · 2025-11-01

**Soundness:** 2
**Presentation:** 3
**Contribution:** 2
**Rating:** 6
**Confidence:** 2

**Summary:**

### **Paper Summary and Key Contributions**
This paper addresses the challenge of **Cross-Architecture Knowledge Distillation (CAKD)**, aiming to transfer global contextual knowledge from Transformers (strong at long-range dependency modeling but computationally heavy) to CNNs (efficient with excellent local feature modeling, dominant in industrial applications). Existing CAKD methods either fail to resolve inductive bias discrepancies between the two architectures or reduce informative features to logits (limiting generalization across tasks) .  To overcome these issues, the authors propose a **feature-based CAKD framework**. It's has three key contributions: 1. Proposes the **GIS module** to capture global supplementary information beyond local CNN features, helping CNN students better absorb knowledge from Transformer teachers .  2. Applies **ℓ₁-regularization** to supplementary components to mimic Transformer attention characteristics, realizing semantic information alignment .  3. Achieves state-of-the-art performance on both image classification and instance segmentation tasks, outperforming existing advanced CAKD methods consistently .

**Strengths:**

### **List of Strengths**
1. **Targeted Solution to Existing CAKD Limitations**
   The paper directly addresses the two core drawbacks of current Cross-Architecture Knowledge Distillation (CAKD) methods: failure to resolve inductive bias discrepancies between Transformers (global context-focused) and CNNs (local feature-focused), and over-reliance on logits that weakens downstream task generalization. By proposing a feature-based framework instead of generic or logits-centric strategies, it effectively mitigates these issues and bridges the knowledge transfer gap between the two architectures .

2. **Dual-Perspective Feature Alignment Design**
   The framework achieves thorough alignment of Transformer-CNN representations through two complementary components:
   - **Structural alignment**: The Global Information Supplement (GIS) module (inspired by non-local modules) captures residual cues via global-local comparison, enabling compatible feature transfer for CNNs to absorb Transformer global knowledge .
   - **Semantic alignment**: ℓ₁-regularization enforces sparse global compensation patterns, mimicking the intrinsic attention characteristics of Transformers and ensuring semantic consistency . This dual design ensures both structural compatibility and semantic similarity, rather than one-dimensional optimization.

3. **Strong Practical Applicability**
   The method aligns with industrial needs: it enhances CNN performance (the dominant architecture in industrial applications) using Transformer knowledge, without increasing CNN computational complexity. Additionally, it avoids redundant operations (e.g., proving explicit position encoding is unnecessary for CNNs, as their features already contain spatial information), further optimizing efficiency for real-world deployment .

**Weaknesses:**

### **Weaknesses and Restrictions of the Proposed Method**
1. **Limited Coverage of Teacher-Student Architecture Pairs**
   The method’s validation is restricted to a narrow range of teacher and student models, lacking exploration of more diverse architecture combinations. For teacher models, only small/medium-sized Transformers (Swin-T, ViT-S, DeiT-T) are used, while larger Transformer variants (e.g., ViT-B/16, Swin-B) or domain-specific Transformers (e.g., medical image-focused Transformer) are not tested. For student models, only lightweight CNNs (ResNet-18, MobileNet-V2, ResNet-50) are evaluated, excluding more complex CNN architectures (e.g., ResNeXt, EfficientNet, or industrial-specific defect detection CNNs). This limits the generalizability of the method to scenarios with larger teacher models or non-lightweight CNN students .

2. **Unaddressed Computational Overhead of the GIS Module**
   The proposed Global Information Supplement (GIS) module is inspired by non-local modules, which are inherently associated with high computational complexity . However, the paper does not provide quantitative analysis of the GIS module’s computational cost compared to baseline CAKD methods or vanilla CNNs. For industrial applications sensitive to real-time performance, unquantified computational overhead may become a critical barrier to deployment .

3. **Lack of Analysis on Hyperparameter Sensitivity**
   The method relies on key hyper-parameters (e.g., the weight β of the distillation loss, the strength of ℓ₁-regularization, and parameters of the GIS module’s linear transformation W₁) to achieve optimal performance. However, the paper does not conduct a systematic sensitivity analysis. This increases the practical application cost, as users may need extensive tuning to adapt the method to their specific scenarios .

4. **Restricted Exploration of Distillation Feature Layers**
   The paper specifies that teacher and student features are extracted from the **penultimate layer** for alignment, but does not explore the impact of distilling features from other layers (e.g., shallow layers for low-level texture information, middle layers for semantic cues). It remains unclear whether the method’s effectiveness is limited to the penultimate layer, or if multi-layer distillation (combining features from different layers) could further improve performance. This one-dimensional layer selection restricts the method’s flexibility in adapting to different architecture characteristics .

**Questions:**

### **Questions and Suggestions for  Authors**

1. **How Does the Method Perform Under Small-Sample or Imbalanced Data Scenarios?**
All experiments use well-annotated, large-scale datasets with no testing on small-sample or class-imbalanced data. Since industrial scenarios often lack sufficient labeled data, it is unclear whether the method’s feature distillation can still bridge Transformer-CNN gaps when training data is limited.

2. **Can the Method Be Combined With Data-Free Knowledge Distillation?**
The paper focuses on data-dependent CAKD but does not discuss compatibility with data-free distillation. Given that privacy-preserving AI is critical in industries like healthcare, it is unknown whether the GIS module and ℓ₁-regularization can be adapted to data-free settings.

3. **Add Ablation Studies on Feature Layer Selection**
The paper uses only the penultimate layer for feature alignment . Conduct ablation studies on distilling features from shallow (low-level texture) or middle (semantic) layers to determine if multi-layer distillation (combining penultimate + middle layers) can further improve performance. This would clarify the optimal layer choice for different architectures.

---

> ### Author Response · Authors · 2025-11-21
> **Response to Reviewer i5NX**
>
> **Response to W1:** For a fair comparison, we follow the base protocol established by prior works. This inevitably restricts the set of model combinations we can compare. Introducing new combinations would require re-evaluating the accuracy of all prior methods, which is prohibitively time-consuming. Moreover, using a larger teacher architecture introduces another well-known challenge in knowledge distillation: a stronger model does not necessarily make a better teacher. This may undermine the meaningfulness of the final accuracy comparison.
>
> ---
>
> **Response to W2:** Since the GIS also plays the role for aligning the dimensions between teacher and student models, the extra parameters and computational costs of GIS heavily rely on the architectures of T and S. To more intuitively demonstrate the complexity of our method, we compare the extra parameters and the overall distillation FLOPs introduced by our method with those of the OFA. The results are shown in the table below, where the FLOPs are computed using the ''thop.profile'' function. The extra parameters and overall FLOPs are similar and even lower than OFA.
>
> | Dataset | Swin-T Res-18 | ViT-S Res-18 | Swin-T MV-2 | ViT-S MV-2 | Swin-T Res-18 | DeiT-T Res-18 | Swin-T MV-2 | DeiT-T MV-2 |
> |---------|----------------|--------------|--------------|-------------|----------------|----------------|--------------|--------------|
> |        | CIFAR-100  |              |              |             | ImageNet-1K|                |              |              |
> | FLOPs of OURs  | 6281.03 M | 6169.07 M | 4764.37 M | 4648.79 M | 6282.18 M | 2931.38 M | 4766.21 M | 1409.99 M |
> | Extra Params of OURs | 1.76M | 0.53M | 1.91M | 0.61M | 1.76M | 0.16M | 1.91M | 0.20M |
> | FLOPs of OFA | 6286.19 M | 6147.11 M | 4774.43 M | 4651.71 M | 6290.11 M | 2979.61 M | 4780.88 M | 1487.66 M |
> | Extra Params of OFA | 1.63M | 1.19M | 2.53M | 2.53M | 4.39M | 3.04M | 7.14M | 7.14M |
> | Params of S | 11.23M | 11.23M | 2.35M | 2.35M | 11.69M | 11.69M | 3.50M | 3.50M |
>
> ---
>
> **Response to W3:** We apologize for not specifying the hyperparameter settings clearly in the paper, which may have caused confusion. We performed a grid search for β, while λ was used purely as a regularization coefficient and fixed to 1e-4. Taking Swin-T & Resnet8 as an example, we observed that the performance becomes stable once \bate exceeds a certain range as follows.
>
> | k5   | k10  | k15  | k20  | k25  | k30  | k35  |
> |------|------|------|------|------|------|------|
> | 82.12 | 83.62 | 83.97 | 84.11 | 84.27 | 84.31 | 84.27 |
>
> Therefore, we set β=30 for the experiments about Swin-T models in our paper, while for other configurations like the reverse-direction distillation experiment from a CNN to a Transformer-like model that we will add (as below), β is set to 1.
>
>
> | Student       | Teacher | OFA   | RSD | LDRLD | PAT | Ours  |
> |---------------|---------|-------|--------|----------|--------|-------|
> | ConvNeXt-T    | Swin-P  | 78.32 | 82.21  | 80.71    | 80.74  | **82.41** |
> | ConvNeXt-T    | DeiT-T  | 75.76 | 82.46  | 77.46    | 79.59  | **83.09** |
>
> ---
>
> **Response to W4:** Based on our previous study experience, for image classification—a relatively simple task—distilling the penultimate features (right before the avgpool layer) typically yields the best effectiveness. As shown in the table, using the same hyperparameters, distilling intermediate features leads to weaker performance due to insufficient constraint strength, while multi-layer distillation introduces additional parameters but offers no clear improvement. Therefore, we distill only the penultimate features rather than incorporating other layers, as our method is not specifically designed for multi-layer feature distillation like ReviewKD. In contrast, for more complex dense prediction tasks, e.g., object detection and instance segmentation, distilling the backbone output features is likely to yield the best performance.
>
> | Student | Intermediate Layer | Penultimate Layer | Multi-Layer |
> |---------|--------------|-------------|------------|
> | 74.01   | 76.89        | 84.31       | 84.23      |
>
> ---
>
> **Response to Q1:** Our method targets the standard large-scale setting used in prior distillation work, and its architecture-level feature alignment does not rely on dense or balanced data distributions. Due to space and scope limitations, specialized low-data experiments are left for future work.
>
> ---
>
> **Response to Q2:** Our method is designed for data-dependent KD. Since data-free sample generation is an orthogonal research direction, integrating our method into DF-KD pipelines is promising but beyond the scope of this paper.
>
> ---
>
> **Response to Q3:** Please refer to the Response to W4.

---

### Note · Authors · 2026-01-26

I have read and agree with the venue's withdrawal policy on behalf of myself and my co-authors.

---

### Meta-Review · Area_Chair_jMNe · 2026-01-07

**Summary:**

This paper explores Cross-Architecture Knowledge Distillation (CAKD), specifically targeting the transfer of global contextual knowledge from Transformer teachers to CNN students. The authors propose a "Global Information Supplement" (GIS) module, which uses an attention-style mechanism to aggregate global features for the CNN, and an L1-regularization term to enforce sparsity in these supplementary patterns. While the reviewers appreciate the motivation to bridge the inductive bias gap between Transformers (global) and CNNs (local), the overall consensus leans toward the method being an incremental refinement of existing attention-based distillation techniques. The primary concerns revolve around the limited technical novelty of the GIS module and the marginal empirical improvements, which do not sufficiently demonstrate a breakthrough in handling architectural heterogeneity.

**Reviewer Concerns:**

Addressed by Rebuttal:

- Computational Overhead (i5NX, hshi, 7XXQ): The authors provided a FLOPs and parameter count table, demonstrating that the GIS module’s training overhead is comparable to existing methods like OFA.

- Baseline Comparisons (hshi, 7XXQ): The authors included several ICCV 2025 baselines (RSD, LDRLD, PAT).

- Bidirectional Transfer (7XXQ): Preliminary results were provided for CNN-to-Transformer distillation, showing that the GIS module is not strictly limited to one direction.

Outstanding Part:

- Marginal Empirical Significance: This is the most significant concern. Even with the addition of 2025 baselines, the improvements on ImageNet-1K are often as low as 0.1% - 0.3%. In the highly saturated field of vision distillation, such marginal gains are insufficient to prove the superiority of the proposed "alignment" strategy over other methods.

- Incremental Technical Novelty: The GIS module is essentially a variation of a non-local block or a simplified self-attention layer. Adding an attention layer to a CNN to help it "see" like a Transformer is a well-established concept. The addition of L1-regularization to induce sparsity is a common heuristic and does not constitute a major algorithmic breakthrough. And for the feature selection (i5NX, hshi) part, the authors’ justification for distilling only the penultimate layer, citing "previous experience", is scientifically underwhelming. It fails to address the fundamental discrepancy in hierarchical feature hierarchies between CNNs and Transformers, treating the architecture-level alignment as a single-layer mapping problem.

- Hyperparameter Sensitivity (ZMxK, i5NX): The grid search results for beta show that the model's performance is highly sensitive to distillation weights. This suggests that the reported gains may be more a result of extensive hyperparameter tuning on specific benchmarks rather than a robust, generalizable alignment mechanism.

**Reviewer Scores:**

Reviewer i5NX (Initial: 6): While the overhead concern was answered, the reviewer’s significant critique regarding "restricted exploration of feature layers" was met with a dismissive "experience-based" response.

Reviewer ZMxK (Initial: 6): The rebuttal did not provide the requested "theoretical analysis of convergence" in a meaningful way (only citing general KL-rules), nor did it expand the dataset variety beyond standard classification/segmentation.

Reviewer hshi (Initial: 6): This reviewer specifically noted that the "global vs. local" bottleneck is already explored in several recent works. The narrow margins (<0.3) against the 2025 baselines provided in the rebuttal are unlikely to convince them of the paper's high impact.

Reviewer 7XXQ (Initial: 4): The author addressed the symbol/equation errors and provided reverse-distillation results. However, the reviewer’s core concern about whether this method offers anything truly better than simple logit-based KD remains largely unanswered, as the gains from adding feature alignment are minimal.

Reason for reject: The proposed GIS module is a straightforward application of existing attention modules, and the semantic alignment via L1 is an incremental addition. Most importantly, the empirical "wins" are too narrow and marginal to prove that this method is a significant advancement over current SOTA. The rebuttal failed to resolve the concern that the improvements are primarily due to hyperparameter optimization rather than a superior alignment strategy.

---

### Decision · Program_Chairs · 2026-01-26

Reject